# Incidence and Predictors of Acute Kidney Injury Following Advanced Ovarian Cancer Cytoreduction at a Tertiary UK Centre: An Exploratory Analysis and Insights from Explainable Artificial Intelligence

**DOI:** 10.3390/curroncol32020073

**Published:** 2025-01-28

**Authors:** Elizabeth Ratcliffe, Ciara Devlin, Sarika Munot, Timothy Broadhead, Amudha Thangavelu, Michela Quaranta, David Nugent, Evangelos Kalampokis, Diederick De Jong, Alexandros Laios

**Affiliations:** 1School of Medicine, University of Leeds, Leeds LS2 9LU, UK; um21err@leeds.ac.uk (E.R.); um19cmd@leeds.ac.uk (C.D.); 2Department of Gynaecologic Oncology, ESGO Centre of Excellence for Ovarian Cancer Surgery, St James’s University Hospital, Leeds LS9 7TF, UK; s.munot@nhs.net (S.M.); tim.broadhead@nhs.net (T.B.); amudhathangavelu@nhs.net (A.T.); michela.quaranta@nhs.net (M.Q.); david.nugent@nhs.net (D.N.); diederick.dejong@nhs.net (D.D.J.); 3Information Systems Lab, Department of Business Administration, University of Macedonia, 54636 Thessaloniki, Greece; ekal@uom.edu.gr

**Keywords:** acute kidney injury, ovarian cancer, surgical cytoreduction, artificial intelligence

## Abstract

Background/Objectives: The incidence of acute kidney injury (AKI) following advanced epithelial ovarian cancer (EOC) surgery has not been extensively studied. This study aimed to investigate the incidence of AKI and identify preoperative and intraoperative predictors in patients undergoing advanced EOC cytoreduction using both traditional statistics and Artificial Intelligence (AI) modelling. Methods: Retrospective data were collected for 134 patients with a suspected or confirmed diagnosis of advanced EOC (FIGO Stage III–IV) who underwent surgical cytoreduction between January 2021 and December 2022 at a UK tertiary referral centre. AKI was diagnosed according to the KDIGO criteria. Data on 22 patient variables were extracted, including age, Charlson Comorbidity Index (CCI), procedure length, surgical complexity, and length of hospital stay. Logistic regression analysis was used for feature selection to identify AKI predictors, and an extreme gradient boost (XGBoost) model was applied to all variables related to AKI events. Results: The incidence of postoperative AKI was 6.72% (n=9). Predictive factors for AKI included younger age (OR = 0.942, p=0.037), lower CCI (OR = 0.415, p=0.015), longer procedure duration (OR = 1.006, p=0.019), and greater surgical effort (OR = 1.427, p=0.007). Patients with perioperative AKI experienced a doubling in the length of hospital stay (p=0.008). Mortality rates were similar between patients with and without AKI. AI-driven algorithms highlighted the complexity of AKI prediction and provided individual risk profiles, enabling future stratification and prompting different frequencies of AKI monitoring following cytoreduction. Conclusions: Predicting AKI is a complex task. This study found a lower-than-expected incidence of AKI following advanced EOC cytoreductive surgery. AKI is linked to heightened surgical risk-taking, underscoring the need for improved guidelines focusing on postoperative monitoring for targeted patients. Artificial Intelligence offers the potential for personalized AKI prediction.

## 1. Introduction

Epithelial ovarian cancer (EOC) is often diagnosed at advanced stages [1]. The cornerstone of EOC treatment is a combination of cytoreductive surgery (CRS) and platinum-based chemotherapy. Recent findings strongly indicate that achieving no macroscopic residual disease (CC 0) in advanced EOC leads to longer overall survival rates [2]. Maximal effort cytoreductive surgeries aiming for CC 0 resections in these patients are frequently radical and extensive, carrying a high risk of complications [3].

Acute Kidney Injury (AKI) is defined as a rapid (hours to days) reduction in kidney function. It is a common complication in hospitalized patients, with a reported incidence as high as 18% [4,5]. This risk is associated with increasing age, perioperative nephrotoxic drug use, and a greater comorbidity burden, including diabetes mellitus, liver disease, and heart failure [5]. AKI substantially increases the risk of patient morbidity and mortality, prolongs hospital admissions, and can trigger severe complications, including hyperkalemia, pulmonary edema, metabolic acidosis, and chronic kidney disease (CKD). At its most severe, AKI can result in multi-organ failure, with mortality rates reported as high as 80% [6]. Despite its severity, hospital-acquired AKI has historically been poorly managed. The 2009 National Confidential Enquiry into Patient Outcome and Death (NCEPOD) found that 43% of patients had delayed recognition of AKI, 21% of cases were avoidable, and only 50% of patients received good care [4]. Given the high incidence and delays in prompt management, AKI costs NHS England approximately GBP 1.02 billion per year [6]. It is a costly complication for both the patient and the healthcare system.

Intraperitoneal surgery is another significant risk factor for AKI development, as EOC’s laparotomy approach and radical nature can result in a high risk of significant blood loss, renal damage, and infection, thereby increasing the likelihood of AKI [7,8]. This is especially pertinent in the management of advanced EOC, where CRS aims to resect all macroscopic disease alongside adjuvant or neoadjuvant chemotherapy [9,10]. The incidence of AKI post-CRS for EOC widely varies in the literature. A single-centre retrospective cohort study reported that among 282 patients undergoing CRS, 11.7% developed AKI [5]. This incidence was higher in patients with hypertension or diabetes mellitus; ACE inhibitor or ARB use, and a baseline eGFR <60 were independent risk factors for AKI. In contrast, another single-centre retrospective study found a much higher incidence of AKI [7]. Among 47 patients who underwent CRS, 40.4% developed AKI, with significant risk factors being age >48 years, baseline eGFR <90, the interval between neoadjuvant chemotherapy and the operation date <7 days, and intraoperative blood transfusion >2 units. Other studies identified Charlson Comorbidity Index (CCI), platinum-based chemotherapy, and high estimated blood loss (EBL) as significant predictors of AKI in the context of CRS [11,12].

Despite the well-documented risks of AKI post-surgery, existing literature often focuses on AKI secondary to CRS with hyperthermic intraperitoneal chemotherapy (HIPEC), which does not align with NICE clinical guidelines [9] and presents a confounding factor given the nephrotoxic nature of platinum-based chemotherapeutic agents [8]. The primary aim of this study was to investigate the incidence of AKI in patients undergoing CRS without HIPEC for advanced EOC and to perform an exploratory analysis of preoperative and intraoperative predictors of AKI in these patients. Secondary aims included using AI modelling to enhance data analysis and assess whether AKI affected patients’ postoperative outcomes. The study’s endpoints included AKI incidence, predictors of AKI, and secondary outcomes influenced by AKI.

## 2. Materials and Methods

### 2.1. Study Population

Retrospective data were collected for 138 patients with a confirmed or working diagnosis of advanced EOC (FIGO Stage III–IV) who underwent CRS between January 2021 and December 2022 at a UK tertiary referral centre, which is an ESGO accredited centre of excellence for ovarian cancer surgery. Inclusion criteria required patients to be 18 years or older with a diagnosis of advanced EOC by CT thorax/abdomen/pelvis and/or histological confirmation by omental biopsy. They were also included irrespective of their chronic kidney disease (CKD) status. Exclusion criteria included non-epithelial histology, synchronous non-ovarian primary tumours, and early-stage or borderline EOC. Patients were excluded if there was no attempt at resection due to inoperable disease. Before making treatment decisions, all cases were discussed at the central gynaecological oncology multidisciplinary meeting.

### 2.2. Study Design and Data Collection

The study was a single-centre, retrospective, exploratory cohort review of clinical case notes. The patient cohort was identified retrospectively by their CRS operative record, and data were extracted from various sources within the electronic health records (EHRs), including their preoperative assessment, operation note and discharge summary. Data on patient’s preoperative characteristics including age, body mass index (BMI) and CCI, as well as intraoperative factors including procedure length (PL), EBL, the ANAFI score [13], and Aletti Surgical Complexity Score (SCS) were collected (Table A1). Additionally, postoperative data were collected for secondary outcomes which aimed to illustrate the impact AKI had on patients and NHS workload; these included postoperative length of stay (LOS), HDU/ICU admission and short-term mortality within 90 days of the procedure. The study was conducted in accordance with the guidelines of the Declaration of Helsinki and was approved by the Institutional Review Board (23/NE/0229/328779/12.01.24). It was registered in the UMIN/CTR Trial Registry under the identifier UMIN000049480.

### 2.3. Identification and Classification of AKI

The Kidney Disease: Improving Global Outcomes (KDIGO) definition and staging criteria were used to identify AKI in this study, aligning with institutional AKI guidelines [6] and NICE Guideline NG148 [4]. KDIGO defines AKI as a rise in serum creatinine of 26.5 µmol/L or more within 48 h, an increase in serum creatinine of 50% within one week, or urine output less than 0.5 mL/kg/hour for six consecutive hours [14].

Patients whose biochemistry results met KDIGO criteria were automatically flagged for AKI in the EHR system, with staging calculated via an algorithm in the Lab Information Management System (LIMS) based on KDIGO definitions (see Table A2) [6]. In perioperative care, serum creatinine is routinely measured pre- and postoperatively, providing reliable and objective data. If preoperative serum creatinine was not recorded, the most recent value before surgery was used as the baseline. Measurement of urinary output to identify AKI incidence was excluded in the planning phase due to the prediction of missingness in retrospective data collection from case notes. If an AKI alert was triggered, clinical teams implemented the STOP AKI care bundle to confirm or refute the clinical diagnosis of AKI. Patients with end-stage kidney disease on dialysis trigger the alert and patients who have been commenced on diuretics or ACEi/ARBs for heart failure may experience a rise in creatinine that is acceptable but not clinically AKI. Clinical data are routinely collected, analyzed, and reviewed at a specialty level to recommend service changes where appropriate.

### 2.4. Statistical Analysis

Microsoft Excel was used to calculate the percentage incidence of AKI. For preoperative, intraoperative and postoperative variables, numerical data were expressed using medians and categorical data were expressed using percentages. SPSS V24^®^ was used to perform Mann–Whitney U tests to identify differences in preoperative and intraoperative variables and postoperative secondary outcomes between the AKI and non-AKI groups. Spearman’s Rank tests identified significant associations between preoperative, intraoperative and postoperative variables, and correlation heatmaps were generated. Univariate logistic regression analyses identified important predictors of AKI. A *p*-value of <0.05 was considered to indicate statistical significance.

The extreme gradient boost (XGBoost, Python Package—xgboost 2.1.3) classifier was employed to model all variables pertaining to AKI events [15]. Receiver operating characteristic (ROC) curves were used to test model performance. Explainable Artificial Intelligence (XAI) was used to explain the prediction, and SHAP Force Plots were generated to illustrate an individual patient’s feature importance and to identify their protective and risk factors to calculate their probability of developing an AKI [13]. In that same study, we identified a novel intraoperative score (ANAFI) threshold equal to 8, below which it is more likely to achieve complete cytoreduction [13]. Assuming a 50% difference in AKI incidence between the ANAFI < 8 and ANAFI > 8 groups, a minimum sample size of 140 patients would have been required to detect the difference with a probability of type 1 error α=0.05, and a probability of type 2 error β=0.2 for a study power of 0.8. Our study cohort was close to the required number to provide some meaningful conclusions.

## 3. Results

A total of 138 patients were listed for CRS for advanced EOC in Leeds Teaching Hospitals Trust in the years 2021–2022. Four patients were excluded from data analysis as no attempt at resection was made. Of the 134 patients included in the study, the total incidence of AKI post-CRS was 6.72% (n=9). Of the AKI patients, 88.9% (n=8) had KDIGO AKI Stage 1, no patients had Stage 2 and 11.1% (n=1) had Stage 3. Overall, 44.4% of patients (n=4) with AKI had a pre-renal cause (hypovolaemia n=3, infection n=1) and 11.1% (n=1) had a renal cause (nephrectomy n=1). In addition, 44.4% (n=4) had no suggested cause of AKI given in their discharge summaries.

With regards to preoperative patient characteristics (Table 1), the cohort median age was 65.5 years. The AKI group had a younger median age of 55 years compared to the non-AKI group’s median age of 66 years (p=0.018). The median BMI was 29.7 in the AKI group and 25.85 in the non-AKI (p = 0.102). The CCI was significantly lower in the AKI patients (n=7) versus non-AKI patients (n=8) (p=0.008). Seven patients had a diagnosis of CKD, and none of them went on to develop a postoperative AKI.

There were no significant differences in the percentage of patients using ACE inhibitors, ARBs, NSAIDs or diuretics who developed an AKI compared to those who did not. In total, 33.3% of AKI patients reported being regular smokers in their preoperative assessment compared to 12.8% in non-AKI patients, although this was not a significant difference (p=0.079). The median Eastern Cooperative Oncology Group Performance Score (ECOG PS) was zero in both groups, indicating a normal functional baseline across the cohort before surgery. The median baseline eGFR was >90 in the non-AKI group and 88 in the AKI group (p=0.079). Differences between baseline creatinine and baseline albumin were negligible. The percentage of patients undergoing neoadjuvant chemotherapy was similar (55.6% vs. 52%, p=0.905).

Of the intraoperative variables (Table 2), there were marked differences in procedure length and surgical complexity. The AKI group underwent longer surgeries with a median length of 255 min compared to the non-AKI group’s median of 205 min (p=0.033). The AKI median SCS was seven compared to the non-AKI median SCS of 3 (p=0.034). There was no difference between the intraoperative fluid statuses of either group, including EBL and fluid volume given. In terms of surgical outcomes, 72.4% (n=97) of patients had a CC 0 achieved. Overall, 17.9% (n=24) had a CC 1, meaning that residual nodules were smaller than 2.5 mm, and 9.7% (n=13) had a CC2, meaning residual nodules were between 2.5 mm and 2.5 cm.

Postoperatively, three patients from the non-AKI group died in the 90 days following CRS, although their cause of death was not examined in the scope of this study. The short-term mortality rate following CRS for advanced EOC in this study was 2.2%. A total of 20.1% (n=27) required more complex medical support and were admitted to HDU/ICU following surgery. The rate of HDU/ICU admission was 33.3% in the AKI group compared to 19.2% in the non-AKI group (p=0.283). Significantly, the AKI group spent twice as long in hospital post-CRS with a median stay of 12 days compared to 6 days for the non-AKI group (p=0.008) (Table 3).

Spearman’s rank correlations between the significant factors showed associations between the preoperative, intraoperative and postoperative outcomes (Table 4). Age was significantly positively correlated with CCI (C=0.889, p<0.001), age being a component of CCI. The Charlson Comorbidity Index was negatively correlated with procedure length (C=−0.183, p=0.034), indicating less comorbid patients received longer operations. Aletti SCS and procedure length were also positively correlated (C=0.532, p<0.001), suggesting that more complex surgeries lasted longer. A longer procedure length also correlated with a longer stay postoperatively (C=0.43, p<0.001).

Univariate logistic regressions (Table 5) for age (OR 0.942, 95% CI 0.891, 0.996) and CCI (OR 0.415, 95% CI 0.205, 0.841) indicated that they were significant predictors for AKI. Baseline eGFR, creatinine and albumin were not significant predictors for AKI, suggesting AKI incidence was not dependent on preoperative renal function. For intraoperative variables, procedure length (OR 1.006, 95% CI 1.001, 1.012, p=0.019) and Aletti SCS (OR 1.427, 95% CI 1.104, 1.844, p=0.007) were both significant predictors of AKI outcome.

Feature importance was calculated in machine learning based on the whole cohort (Figure A1). In line with conventional regression analysis, AI identified that age and SCS were the most important features in determining a patient’s risk of AKI development, although the feature importance plots did not indicate whether a variable was a risk factor or a protective factor. Neoadjuvant chemotherapy, CCI and procedure length were reported as less important features compared with suggestions by traditional statistics. TAI identified additional predictors of AKI compared to SPSS. The feature importance of the thirteen most important AI predictive features is illustrated in a parallel coordinate plot (Figure 1). The XGBoost was superior to conventional regression for AKI prediction (area under curve (AUC) = 0.85 vs. 0.72).

For local explainability, AI-based SHAP Force Plots displayed both risk and protective factors for individual patients and visualised the probability of AKI development for a patient based on their unique characteristics. This was ‘learned’ from the cohort’s characteristics and then applied to one patient.

Figure 2 displays example SHAP Force Plots for a single patient. The blue features decreased AKI risk, and the red features increased the risk. In Plot A, the calculated SHAP value risk was 0.69 due to younger age and higher surgical complexity, which were high-risk factors for the patient, whereas for Plot B, the odds of AKI was 0.61 despite other red features being added because age and surgical complexity had a weaker predictive value for that specific patient, thus lowering the risk.

## 4. Discussion

Our study found a lower-than-expected incidence of AKI following CRS for advanced EOC. The reported incidence ranges widely between 11.4% [5] and 40.4% [7] in the literature. This could reflect the difficulties in accurate and prompt AKI monitoring, first because AKI has multiple diagnostic criteria (KDIGO, RIFLE, AKIN and NCI-CTCAE) [5,7,11] and second because decreasing urine output is harder to accurately measure and report than changing biochemical values, thus making comparisons between studies challenging. In that sense, the exclusion of diagnostic urine output is an accepted limitation of this study that may have resulted in under-reporting AKI incidence. Although the reasons for this discrepancy were not examined in the scope of this study, one such reason could be the absence of HIPEC, a known risk factor for AKI development [5,8]. Furthermore, a meta-analysis reviewing the incidence of AKI following 12,947 major gynaecological surgeries found that 7% of patients with appropriate monitoring developed an AKI [16].

Compared to the current literature, some results from this study contradicted the initial working hypothesis. The preoperative factors expected to be predictive of AKI were increased age, use of an ACEi/ARB, a reduced baseline eGFR, an increased CCI, a low baseline albumin and neoadjuvant chemotherapy. Contrastingly, this study found younger age and decreased CCI to be predictive of AKI. Higher Aletti SCS and increased procedure length were found to be predictive of AKI in this cohort. The correlation between procedure length and increasing SCS and decreasing CCI suggests that healthier patients are selected for more radical surgeries. Longer surgeries increase exposure to potential nephrotoxic factors such as prolonged anaesthesia and intraoperative hypotension resulting in a higher risk of developing a postoperative AKI. As complete cytoreduction is required to ensure the best survival outcomes, maximal effort CRS is likely to be exerted by surgeons in younger patients with fewer comorbidities who are likely to better tolerate the surgical acuity.

Baseline eGFR, creatinine and albumin levels were not found to be predictive of AKI. We suspect that the missing baseline eGFR for five patients and the missing baseline albumin for thirty-three patients, which represents 33.3% of the AKI group, are responsible for this. It is possible that with the missing data, baseline albumin may have been a significant predictor for AKI as suggested in the literature [7]. Additionally, within EHRs all normal kidney function values are recorded as >90 mL/min/1.73 m^2^ to run data analysis, this had to be taken as an eGFR of 90 which limits the accuracy of eGFR in statistical analysis. Meanwhile, AI models could input eGFR >90, and therefore eGFR is possibly more accurately analyzed in the machine learning figures. Overall, our study suggests that patients with a sub-optimal eGFR can still sustain a complex CRS without an increased risk of AKI in the postoperative window.

Neoadjuvant chemotherapy was not found to be a predictive risk or protective factor for AKI development by regression analysis, which suggests that fears of chemotherapy-related nephrotoxicity contributing to AKI should not interfere with surgical planning when it comes to CRS. Chemotherapeutic agents and neoadjuvant regime timings and their association with completeness of resection was not assessed in the scope of this study. Blood loss was not a predictor of AKI in this study, despite hypovolaemia being the modal stated cause of AKI in discharge summaries. This is likely because the EBL and fluid balance during surgery were monitored and carefully corrected by the anaesthetic team and the hypovolaemia was likely to have occurred in the early postoperative period due to the fluid shift. Causes of postoperative hypovolaemia leading to AKI were not explored in this study.

Patients with AKI experienced a significantly longer median length of hospital admission of 12 days compared to 6 days for non-AKI patients, akin to the post-AKI length of stay across other disciplines [6]. We previously demonstrated that maximal effort at CRS comes at the expense of significantly increased LOS by two days [17]. Patients with AKI also had a greater need for intensive or high-dependency care (33.33% vs. 19.20%) as several common factors could be shared between HDU admission and AKI predictions [18]. Notably, while three patients in the non-AKI group died within 90 days of admission, there were no fatalities amongst the AKI group.

Despite multiple factors risking renal function during CRS for advanced EOC, the study adds to the scarce literature regarding AKI incidence, predictors and outcomes in patients receiving CRS without HIPEC. The limitations of a single-centre retrospective study need to be acknowledged. However, the study provides meaningful insights into postoperative AKI development as a result of the primary management of EOC within the UK. This is increasingly important given the projected rise in ovarian cancer incidence of 5% by 2038 [19]. Additionally, the strength of AI-guided data interpretation over traditional statistics allowed the identification of trends in a whole cohort. We were able to make predictive insights for patients on a population or individual level.

Most patients experienced Stage 1 AKI. Rapid reductions in kidney function were automatically flagged in the EHRs, leading to better utilisation of the AKI bundle. The renal team was appropriately involved in all Stage 3 AKI cases. However, event documentation was minimal in discharge letters. One Stage 3 AKI case was caused by a known ureteric leak and small bowel ischemia during cytoreductive surgery. Arguing that creatinine can be reabsorbed from a ureteric leak in the peritoneal cavity, this occurred as an undesirable event from CRS, and this patient met the diagnostic criteria for an AKI, hence their inclusion. Similarly, while a small rise in creatinine is expected following a nephrectomy, this rise was automatically flagged for AKI in the HER. By meeting all the inclusion and AKI diagnostic criteria, the nephrectomy patient was still included in this study. Strictly speaking, one could argue that such cases should not be included, which would further support the low AKI incidence figures at our centre. Notably, the STOP AKI bundle (Table A3) was not completed in any of the AKI cases, but hypovolemia was effectively managed according to protocol.

Our results demonstrated an unexpected AKI risk profile. While we cannot recommend changes to CRS procedures to reduce AKI incidence—since survival benefits potentially outweigh the perioperative risks—this research can support improved postoperative guidelines and enhance clinical education. Resident doctors must complete the STOP AKI bundle during daily rounds and document AKI events at discharge to inform GPs. Staff should distribute appropriate patient information leaflets. Advanced EOC patients present unique postoperative challenges, requiring meticulous daily documentation of fluid management plans for those with AKI. Effective management of AKI risk requires interdisciplinary collaboration. Explainable AI could help introduce a formal AKI risk assessment tool into the postoperative guideline to encourage more vigilant monitoring and timely intervention. Early recognition could also minimize the length of postoperative admission; even a reduction of one day could save the NHS GBP 2349 per day and improve patient outcomes [20].

There is scope for further research assessing the clinical utility of AI models in predicting AKI and the potential for the development of real-time, point-of-care online tools to enable clinicians to make more informed decisions based on a patient’s individual risk assessment.

## 5. Conclusions

Prediction of AKI is a complex task. The study found a lower-than-expected AKI incidence following advanced EOC cytoreductive surgery. AKI results from a multifactorial interplay of patient-specific and surgical factors. It is linked to heightened surgical risk-taking, highlighting the scope for improved guidelines focusing on postoperative monitoring for targeted patients. Artificial intelligence offers the potential for personalized AKI prediction in cytoreduced patients.

## Figures and Tables

**Figure 1 curroncol-32-00073-f001:**
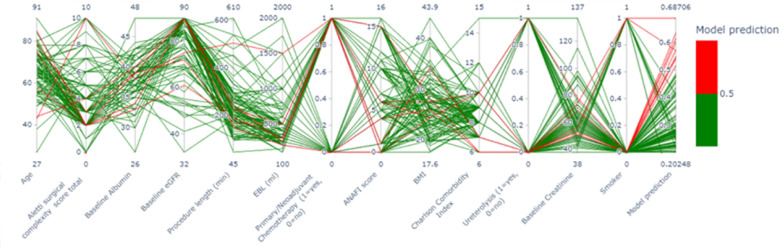
Parallel Coordinate Plot generated by XGBoost Classifier to illustrate the thirteen most important preoperative and intraoperative variables for AKI prediction. Lines represent individual patients, the relationships between their variables and whether they correspond to the prediction of an AKI for that patient (red lines represent AKI predicted, green lines represent no AKI predicted). Red = yes, Green = no. EBL = Estimated Blood Loss, ANAFI = Anatomic Fingerprints Score for the Prediction of Complete Cytoreduction, BMI = Body Mass Index.

**Figure 2 curroncol-32-00073-f002:**
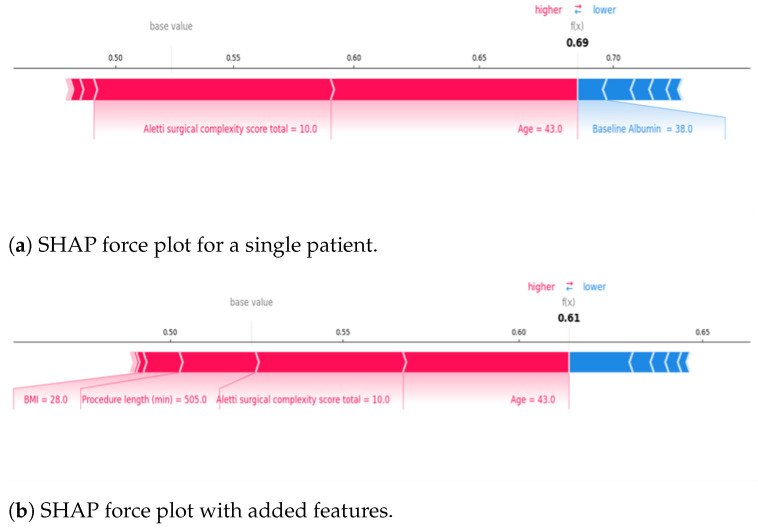
SHAP force plots generated by the AI-driven XGBoost Classifier to illustrate AKI risk scores for (**a**) a single patient and (**b**) the same patient with additional important features for patient-specific AKI prediction. The combined feature impact is the predicted AKI risk. The patient’s predicted risk for AKI was 0.69, driven by younger age, high surgical effort and normal albumin. The risk decreased to 0.61 when BMI of 28 and procedure time of 505 min were factored in.

**Table 1 curroncol-32-00073-t001:** Descriptive statistics for the preoperative variables, comparing the AKI and non-AKI groups, supported with the *p*-value obtained from Mann–Whitney U tests. Significant *p*-values are marked with *.

Characteristic	All (n = 134)	AKI Group (n = 9)	Non-AKI Group (n = 125)	*p*-Value
**Pre-Operative Factors**				
Age (years)	65.5 (58.25, 72)	55 (52, 65)	66 (59, 73)	0.018 *
BMI (kg/m^2^)	26 (22.85, 29.7)	29.7 (27.9, 33.1)	25.85 (22.83, 29.58)	0.102
CCI	8 (7, 9)	7 (7, 8)	8 (8, 9)	0.008 *
CKD	7 (5.2%)	0 (0%)	7 (5.6%)	0.468
ACEi/ARB	26 (19.4%)	1 (11.1%)	25 (20%)	0.510
Diuretic	7 (5.2%)	0 (0%)	7 (5.6%)	0.443
NSAID	10 (7.5%)	1 (11.1%)	9 (7.2%)	0.720
Smoking	19 (14.2%)	3 (33.3%)	16 (12.8%)	0.079
ECOG PS	0 (0, 1)	0 (0, 1)	0 (0, 1)	0.719
FIGO Stage	3 (3, 3)	3 (3, 4)	3 (3, 3)	0.533
Baseline eGFR (mL/min/1.73 m^2^)	90 (77, 90)	88 (80.25, 90)	90 (77, 90)	0.863
Baseline Creatinine (µmol/L)	59 (51, 67)	60.5 (56, 67.75)	59 (51, 67)	0.726
Baseline Albumin (g/L)	38 (36, 40)	38 (34.25, 40.25)	37 (36, 39.5)	0.983
Neoadjuvant chemotherapy	70 (52.2%)	5 (55.6%)	65 (52%)	0.905

BMI = Body Mass Index, CCI = Charlson Comorbidity Index, CKD = Chronic Kidney Disease, ACEi/ARB = Use of an Angiotensin-Converting Enzyme Inhibitor or Angiotensin Receptor Blocker, NSAID = Non-Steroidal Anti-Inflammatory Drug, ECOG PS = Eastern Cooperative Oncology Group Performance Status, FIGO = International Federation of Gynaecology and Obstetrics, eGFR = estimated Glomerular Filtration Rate.

**Table 2 curroncol-32-00073-t002:** Descriptive statistics for the intraoperative variables, comparing the AKI and non-AKI groups, supported with the *p*-value obtained from Mann–Whitney U tests. Significant *p*-values are marked with *.

Characteristic	All (n = 134)	AKI Group (n = 9)	Non-AKI Group (n = 125)	*p*-Value
**Intra-Operative Factors**				
Procedure Length (mins)	210 (171.25, 273.75)	255 (210, 315)	205 (165, 270)	0.033 *
Aletti SCS	3 (2, 5)	7 (3, 8)	3 (2, 4)	0.034 *
EBL (mls)	500 (300, 600)	500 (500, 700)	500 (300, 500)	0.108
Fluid given (mls)	3000 (2000, 4000)	4500 (2813, 5250)	3000 (2000, 4000)	0.202
ANAFI	0 (0, 1)	0 (0, 0)	0 (0, 1)	0.605
Ureterolysis	32 (23.9%)	3 (33.3%)	29 (23.2%)	0.457

Aletti SCS = Aletti Surgical Complexity Score, EBL = Estimated Blood Loss, ANAFI = Anatomic Fingerprints Score for the Prediction of Complete Cytoreduction.

**Table 3 curroncol-32-00073-t003:** Descriptive statistics for the postoperative secondary outcomes, comparing the AKI and non-AKI groups, supported with the *p*-value obtained from Mann–Whitney U tests. Significant *p*-values are marked with *.

Characteristic	All (n = 134)	AKI Group (n = 9)	Non-AKI Group (n = 125)	*p*-Value
**Post-Operative Factors**				
Length of post-operative hospital admission (days)	6 (5, 9)	12 (7, 18)	6 (5, 8)	0.008 *
HDU/ICU Admission	27 (20.1%)	3 (33.3%)	24 (19.2%)	0.283
Hospital Readmission	8 (6%)	1 (11.1%)	7 (5.6%)	0.512
Mortality	3 (2.2%)	0	3 (2.4%)	0.645

HDU/ICU = High-Dependency Unit/Intensive Care Unit, Mortality = Short-term mortality within 90 days of CRS.

**Table 4 curroncol-32-00073-t004:** Results of Spearman’s Rank tests evaluating the significance of correlations between preoperative and intraoperative variables and postoperative outcomes. Significant results are marked with *.

Variable 1	Variable 2	Correlation Coefficient	*p*-Value
Age (years)	Aletti SCS	−0.065	0.458
Age (years)	Procedure Length	−0.161	0.063
Age (years)	Length of Post-operative Admission	0.115	0.187
Age (years)	CCI	0.889 *	<0.001 *
Aletti SCS	Procedure Length	0.532 *	<0.001 *
Aletti SCS	Length of Post-operative Admission	0.389 *	<0.001 *
Aletti SCS	CCI	−0.112	0.200
Procedure Length	Length of Post-operative Admission	0.43 *	<0.001 *
Procedure Length	CCI	−0.183 *	0.034 *
CCI	Length of Post-operative Admission	0.064	0.465

SCS = Surgical Complexity Score, CCI = Charlson Comorbidity Index.

**Table 5 curroncol-32-00073-t005:** Results of the Univariate Logistic regression undertaken on SPSS assessing the predictive value of the preoperative and intraoperative variables. Significant results are marked with *.

Characteristic	Odds Ratio	95% Confidence Intervals	*p*-Value
**Pre-operative Factors**			
Age	0.942	0.891, 0.996	0.037 *
BMI	1.072	0.964, 1.192	0.197
CCI	0.415	0.205, 0.841	0.015 *
ECOG PS	0.919	0.405, 2.083	0.839
FIGO Stage	1.655	0.39, 7.034	0.495
eGFR	1.019	0.954, 1.089	0.567
Creatinine	0.995	0.95, 1.042	0.833
Albumin	0.996	0.795, 1.249	0.975
**Intra-operative Factors**			
Procedure Length	1.006	1.001, 1.012	0.019 *
Aletti SCS	1.427	1.104, 1.844	0.007 *
EBL	1.001	1, 1.003	0.090
Fluid given	1	1, 1.001	0.141
ANAFI	0.627	0.171, 2.302	0.481

BMI = Body Mass Index, CCI = Charlson Comorbidity Index, ECOG PS = Eastern Cooperative Oncology Group Performance Status, FIGO = International Federation of Gynaecology and Obstetrics, eGFR = Estimated Glomerular Filtration Rate, SCS = Surgical Complexity Score, EBL = Estimated Blood Loss, ANAFI = Fingerprints Score for the Prediction of Complete Cytoreduction.

## Data Availability

Data presnted in this study are available on request from the corresponding author.

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
