# Peer review of "Incidence and Predictors of Acute Kidney Injury Following Advanced Ovarian Cancer Cytoreduction at a Tertiary UK Centre: An Exploratory Analysis and Insights from Explainable Artificial Intelligence"

_curroncol, 2025, doi:10.3390/curroncol32020073_

Round 1

Reviewer 1 Report

Comments and Suggestions for Authors

This is a retrospective cohort study analysis, performed in a single tertiary referral institution, aimed at investigating the incidence of acute kidney injury (AKI) and identifying preoperative and intraoperative predictors in patients undergoing advanced epithelial ovarian cancer (FOGO III-IV) cytoreduction without hyperthermic intraperitoneal chemotherapy using both traditional statistics and Artificial Intelligence (AI) modelling.

The paper is well written and the English language is appropriate and understandable.

The clinical topics are interesting. To date, the incidence of AKI following advanced epithelial ovarian cancer (EOC) surgery has not been extensively studied and has frequently been poorly managed.

The Authors found a lower-than-expected incidence of AKI following advanced EOC cytoreductive surgery (6.72%). The literature reported an incidence as high as 18%. Predictive factors for AKI included younger age, higher BMI, lower comorbidity, longer procedure duration, and greater surgical effort. Patients with perioperative AKI have a significantly longer hospitalization.

AI identified that age and surgical complexity score were the most important features in determining a patient’s risk of AKI development.

Unfortunately, the major limitation of this study is the scanty number of patients included in the analysis.

The cited references are mostly recent and include relevant publications.

The conclusions are carefully supported by the results. Prediction of AKI is a complex task and results from a multifactorial interplay of patient-specific and surgical factors. AKI is linked to heightened surgical risk-taking, underscoring the need for improved guidelines focusing on postoperative monitoring for targeted patients. AI offers the potential for personalized AKI prediction.

Specific comments:

Inclusion criteria (rows 75-76) are misleading.

Perioperative patient management for ovarian cancer surgery requires a multidisciplinary approach led by an anesthesiologist and surgeon. Any improvement in peri-operative morbidity, including AKI rates, depends on the success of the multidisciplinary team.

Could the authors confirm the presence of an anesthesiologist team in their referral centre with confirmed experience in the management of gynecologic complex surgical procedures?

Author Response

Comments 1: Inclusion criteria (rows 75-76) are misleading.

Response 1: Thank you for raising this, we agree with this comment. Therefore we have reworded the inclusion criteria to be less vague. "Inclusion criteria required patients to be 18 years or older with a diagnosis of advanced EOC by CT thorax/abdomen/pelvis and/or histological confirmation by omental biopsy". See changes on Page: , Paragraph:, Line: 

Comment 2: 

Perioperative patient management for ovarian cancer surgery requires a multidisciplinary approach led by an anesthesiologist and surgeon. Any improvement in peri-operative morbidity, including AKI rates, depends on the success of the multidisciplinary team.

Could the authors confirm the presence of an anesthesiologist team in their referral centre with confirmed experience in the management of gynecologic complex surgical procedures?

Response 2: Thank you for raising this, the referral centre is a tertiary level teaching hospital with dedicated gynaecology oncology theatres with an experienced anaesthetic team supporting the MDT. 

Reviewer 2 Report

Comments and Suggestions for Authors

The authors present the results of their retrospective review of the incidence and predictors of acute kidney injury in 134 patients undergoing cytoreductive surgery for advanced epithelial ovarian cancer at a single institution.

They concluded, "Prediction of AKI is a complex task. The study found a lower-than-expected AKI incidence following advanced EOC cytoreductive surgery. The AKI results from a multifactorial interplay of patient-specific and surgical factors. It is linked to heightened surgical risk-taking, highlighting the scope for improved guidelines focusing on postoperative monitoring for targeted patients. Artificial Intelligence offers the potential for personalised AKI prediction in cytoreduced patients."

Overall, the research question is important and of contemporary interest. The study was reasonably designed, analyzed and reported.  

I have a number of comments and issues:

1) Lines 26-27: platinum-based chemotherapy for ovarian cancer has not been administered for over a century, let alone as the cornerstone in combination with cytoreductive surgery.  Cis-platinum, the first approved platinum chemotherapeutic agent, was approved in the US in 1978.

2) Lines 45-46: "midline laparotomy can result in prolonged, radical surgery with a high risk of significant blood loss, renal damage, and infection".  I'm not sure what the authors are implying by "can result in" but this sentence is inaccurate and misleading as written.

3) There was no significant association between neoadjuvant chemotherapy and AKI.  However, the authors did not provide any information regarding the patients' neoadjuvant chemotherapy regimen, especially with regards to platinum use (carbo vs. cis), number of cycles, and interval between last cycle and surgery.

4) Lines 136-139: Only need a single decimal point (e.g., 11.1, not 11.11) for these data.

5) Line 138-139: Why include the patient with a nephrectomy?  This clinical scenario is clearly different that patients with pre-renal azotemia.  From a purely definitional perspective, the anticipated rise in creatinine/decrease in GFR immediately following a nephrectomy might meet the criteria for Grade 1 AKI, but it does not have any clinical significance.

6) Lines 272-3: Similarly, the creatinine rises in patients with ureteric leak due to resorption from the peritoneal cavity. Was this truly AKI?  Why include this patient?

7) The authors did not provide any information correlating cytoreductive surgery outcome (e.g., R0, R1, R3) with length of hospital stay. More aggressive surgical procedures to accomplish R0 will, independently prolong hospital stay and increase the probability of HDU/ICU stay.

8) A substantial number of patients were included with missing baseline eGFR and albumin.  The study would be strengthened by re-running and reporting the analysis with and without these patients.

9) No data were provided correlating cytoreductive surgery outcome and neoadjuvant chemotherapy.  One would expect that those patients who received neoadjuvant chemotherapy would require less radical surgery to achieve R0 and, therefore, would be less likely to develop AKI.

10) For the non-significant outcomes, a power analysis would be useful to assess the probability of a Type II error.

11) Define the STOP AKI bundle since it is discussed in the manuscript.

Author Response

Comment 1: Lines 26-27: platinum-based chemotherapy for ovarian cancer has not been administered for over a century, let alone as the cornerstone in combination with cytoreductive surgery.  Cis-platinum, the first approved platinum chemotherapeutic agent, was approved in the US in 1978.

Response 1: Thank you for raising this to our attention, the phrase for over a century was an oversight and we have removed it. "The cornerstone of EOC treatment is a combination of cytoreductive surgery (CRS) and platinum-based chemotherapy."

Comment 2: Lines 45-46: "midline laparotomy can result in prolonged, radical surgery with a high risk of significant blood loss, renal damage, and infection".  I'm not sure what the authors are implying by "can result in" but this sentence is inaccurate and misleading as written.

Response 2: Agree, we have reworded the sentence to not imply that the laparotomy approach was the reason the procedure was radical.

"Intraperitoneal surgery is another significant risk factor for AKI development, as EOC’s laparotomy approach and radical nature can result 45 in a high risk of significant 46 blood loss, renal damage, and infection, thereby increasing the likelihood of AKI [7,8]."

Comment 3: There was no significant association between neoadjuvant chemotherapy and AKI.  However, the authors did not provide any information regarding the patients' neoadjuvant chemotherapy regimen, especially with regards to platinum use (carbo vs. cis), number of cycles, and interval between last cycle and surgery.

Response 3: Thank you for raising this, given the literature regarding the nephrotoxicity of chemotherapy, however we felt the specifics of neoadjuvant treatment fell outside of the scope of this paper as the focus was on CRS and AKI. However we have updated our manuscript to specify that we did not explore the impact chemotherapy had.

"Neoadjuvant chemotherapy was not found to be a predictive risk or protective factor for AKI development by regression analysis which suggests fears of chemotherapy-related nephrotoxicity contributing to AKI should not interfere with 246 surgical planning when it comes to CRS. Chemotherapeutic agents and neoadjuvant regime timings, and their association with completeness of resection was not assessed in the scope of this study."

Comment 4: Lines 136-139: Only need a single decimal point (e.g., 11.1, not 11.11) for these data.

Response 4: Thank you, we have removed the extra digit.

Comment 5: Line 138-139: Why include the patient with a nephrectomy?  This clinical scenario is clearly different that patients with pre-renal azotemia.  From a purely definitional perspective, the anticipated rise in creatinine/decrease in GFR immediately following a nephrectomy might meet the criteria for Grade 1 AKI, but it does not have any clinical significance.

Response 5: By meeting all the inclusion and AKI diagnostic criteria, the nephrectomy patient was still included in this study. For the reason you mentioned, one would argue the nephrectomy case should not be included, which further satisfies our low AKI incidence figures in our centre.

Comment 6: Lines 272-3: Similarly, the creatinine rises in patients with ureteric leak due to resorption from the peritoneal cavity. Was this truly AKI?  Why include this patient?

Response 6: As per response in Q5. By meeting all the inclusion and AKI diagnostic criteria, the creatinine rise, not always assumed in the case of a ureteric leak was automatically flagged up by the system as AKI. Again, it simply confirms the low AKI incidence despite the extensive cytoreductive efforts.

Comment 7: The authors did not provide any information correlating cytoreductive surgery outcome (e.g., R0, R1, R3) with length of hospital stay. More aggressive surgical procedures to accomplish R0 will, independently prolong hospital stay and increase the probability of HDU/ICU stay.

Response 7: Thank you for raising this, although we acknowledge that surgical length and complexity and their correlation to the length of stay is an important secondary outcome of this study. This has been confirmed in a previous study published by our research group (PMID: 37951927). CRS outcome and surgical success were not included in the scope of this study as our primary outcome was AKI development, not surgical success.  The length of hospital stay was examined only as a consequence of AKI.

Comment 8: A substantial number of patients were included with missing baseline eGFR and albumin.  The study would be strengthened by re-running and reporting the analysis with and without these patients.

Response 8: Thank you for highlighting this, we understand that rerunning the study would strengthen our results and conclusions however this isn't possible for us to undertake. Therefore we have acknowledged the impact of the missingness in our discussion and the AI analysis has also taken the missingness into account.

Comment 9: No data were provided correlating cytoreductive surgery outcome and neoadjuvant chemotherapy.  One would expect that those patients who received neoadjuvant chemotherapy would require less radical surgery to achieve R0 and, therefore, would be less likely to develop AKI.

Response 9: Surgical outcomes were not considered as outcomes of this study and therefore the impact of neoadjuvant chemotherapy on surgical complexity and therefore AKI risk is outside our scope. However, it would be an interesting focus for future research - possibly on this data set.

Comment 10: For the non-significant outcomes, a power analysis would be useful to assess the probability of a Type II error.

Response 10: We previously identified an ANAFI score (non-significant) with a cut-off 8 below which complete cytoreduction is more likely to achieve. Assuming a 50% difference in AKI incidence between the ANAFI<8 and ANAFI >8 groups, a minimum sample size of 140 patients would have been required to detect the difference with a probability of type 1 error a=0.05, and a probability of type 2 error beta=0.2 for a study power of 0.8. Our study cohort was close to the required number to provide some meaningful conclusions.

Comment 11: Define the STOP AKI bundle since it is discussed in the manuscript.

Response 11: Thank you for raising this. We agree and will include the bundle poster in our appendices. See Appendix Table A3.

Reviewer 3 Report

Comments and Suggestions for Authors

The manuscript reported a single-centre retrospective study investigating pre-, intra-, and post-operative factors associated with the incidence and outcomes of acute kidney injury (AKI) following ovarian cancer cytoreduction. Patient demographic and clinical data were analyzed using conventional logistic regression analysis and artificial intelligence (AI) model. Data suggested that the increased AKI incidence was associated with decreased patient age and CCI. The authors concluded that AI had the potential for personalized AKI prediction.

Major comments:

1. The sample size was very small with 9 patients in the AKI group. Data from more patients are needed for prediction analysis. Power analysis could be performed to determine proper sample size.

2. The study only analyzed the association between pre-/intra-operative factors and AKI incidence. A cut-off value was not generated for prediction. The false negative/positive rates were not reported. The title and main text need to be revised to accurately reflect the study.

3. ANAFI and ureterolysis results were not described in Results.

4. It is not clear what additional “features were identified to be predictors of AKI by AI” (line 196).

Minor comments:

1. If the p value was greater than 0.05, parameters of two patient groups were comparable (lines 143 and 154).

2. Patients’ racial/ethnic background would be helpful.

Author Response

Comment 1: The sample size was very small with 9 patients in the AKI group. Data from more patients are needed for prediction analysis. Power analysis could be performed to determine proper sample size.

Response 1: Thank for raising this to us. We agree that this is a smaller than the preferred sample size for strong data analysis. Given the limitation of collecting two years worth of data as part of an undergrad project with time constraints, we performed a power analysis based on previous work published by the same group. We previously identified an ANAFI score (non-significant feature) with a cut-off 8, below which complete cytoreduction was more likely to be achievable. Assuming a 50% difference in AKI incidence between the ANAFI<8 and ANAFI >8 groups, a minimum sample size of 140 patients would have been required to detect the difference with a probability of type 1 error a=0.05, and a probability of type 2 error beta=0.2 for a study power of 0.8. Our study cohort was close to the required number to provide some meaningful conclusions, yet this effort remains an exploratory but comprehensive analysis.

Comment 2: The study only analyzed the association between pre-/intra-operative factors and AKI incidence. A cut-off value was not generated for prediction. The false negative/positive rates were not reported. The title and main text need to be revised to accurately reflect the study.

Response 2:

This study was an exploratory analysis of factors pertaining to AKI and there was no need, as clearly stated on the study aim to determine cut-off points in order to translate continuous variables into essentially clinical decisions. I believe the title strongly reflects the effort exerted in this work.

Comment 3: ANAFI and ureterolysis results were not described in Results.

Response 3: Table 2 shows that ANAFI score and ureterolysis were not predictive features. As this was an exploratory analysis, we did not wish to exhaust our results section with describing non-significant results.

Comment 4: It is not clear what additional “features were identified to be predictors of AKI by AI” (line 196).

Response 4: Thank you - we have clarified that the additional features are illustrated in the figure by rewording the sentence. “AI identified additional predictors of AKI compared to SPSS. The feature importance of the thirteen most important AI predictive features is illustrated in a parallel coordinate plot (Figure 1).”

Comment 5: If the p value was greater than 0.05, parameters of two patient groups were comparable (lines 143 and 154).

Response 5: We have adjusted the wording as to not imply significance where there wasn't.

Comment 6: Patients’ racial/ethnic background would be helpful.

Response 6: Although medically relevant and we agree that this could have been an important component for future analysis we did not have access to ethnicity in the areas of the patient record we retrospectively accessed. In fairness, the list of potentially predictive features is already exhaustive.

Round 2

Reviewer 1 Report

Comments and Suggestions for Authors

The  paper can be accepted in present form

Reviewer 3 Report

Comments and Suggestions for Authors

The reviewer's comments have been addressed. There is a typo in line 200. "TAI" should be "AI".